# Predictive Modeling of Vickers Hardness Using Machine Learning Techniques on D2 Steel with Various Treatments

**DOI:** 10.3390/ma17102235

**Published:** 2024-05-09

**Authors:** Claudia Lorena Mambuscay, Carolina Ortega-Portilla, Jeferson Fernando Piamba, Manuel Guillermo Forero

**Affiliations:** 1Semillero Lún, Facultad de Ingeniería, Universidad de Ibagué, Ibagué 730002, Colombia; claudialorena0524@gmail.com (C.L.M.); jeferson.piamba@unibague.edu.co (J.F.P.) 2 Semillero NOVAMAT, Facultad de Ciencias Naturales y Matemáticas, Universidad de Ibagué, Ibagué 730002, Colombia; 2Semillero NOVAMAT, Facultad de Ciencias Naturales y Matemáticas, Universidad de Ibagué, Ibagué 730002, Colombia; 3CONAHCYT-Centro de Ingeniería y Desarrollo Industrial (CIDESI), Santiago de Querétaro 76125, Mexico; iortega@posgrado.cidesi.edu.mx

**Keywords:** Vickers hardness, coating, machine learning, regression, Titanium Niobium Nitride (TiNbN), indentation imprint

## Abstract

Hardness is one of the most crucial mechanical properties, serving as a key indicator of a material’s suitability for specific applications and its resistance to fracturing or deformation under operational conditions. Machine learning techniques have emerged as valuable tools for swiftly and accurately predicting material behavior. In this study, regression methods including decision trees, adaptive boosting, extreme gradient boosting, and random forest were employed to forecast Vickers hardness values based solely on scanned monochromatic images of indentation imprints, eliminating the need for diagonal measurements. The dataset comprised 54 images of D2 steel in various states, including commercial, quenched, tempered, and coated with Titanium Niobium Nitride (TiNbN). Due to the limited number of images, non-deep machine learning techniques were utilized. The Random Forest technique exhibited superior performance, achieving a Root Mean Square Error (RMSE) of 0.95, Mean Absolute Error (MAE) of 0.12, and Coefficient of Determination ($R^{2}$) ≈ 1, surpassing the other methods considered in this study. These results suggest that employing machine learning algorithms for predicting Vickers hardness from scanned images offers a promising avenue for rapid and accurate material assessment, potentially streamlining quality control processes in industrial settings.

## 1. Introduction

Thin film coatings are formed by the controlled deposition of atoms onto the surface of another material, known as the substrate (see Figure 1). This process aims to enhance the substrate’s properties and extend its lifespan. Several factors influence the resulting properties of the coating, including the substrate’s surface roughness, hardness, chemical composition, and material type (ceramic, metallic, polymer, or composite). Additionally, deposition parameters such as nitrogen flow, substrate temperature, deposition time, and thickness play a crucial role [1,2]. Given the multivariable nature of this process, these factors directly impact the coating properties, necessitating careful design and a potentially large number of experiments to fully understand the influence of each parameter.

Therefore, the use of machine learning (ML) techniques helps reduce the number of experiments required to obtain the desired result, such as elemental chemical composition, phases, or specific material properties [3]. ML techniques also enable the classification of material defects, such as fractures or surface stains [4,5,6], or the prediction of coating properties like hardness, friction coefficient, corrosion rate, based on deposition parameters [7,8,9]. One way to control material quality and ensure it meets the appropriate characteristics for its application is through the Vickers hardness test [10]. This test involves measuring the plastic deformation or indentation mark produced on the surface after applying a load with a pyramidal diamond indenter, and the hardness value is determined by Equation (Equation 1).
(1)$$HV=0.1891\frac{F(N)}{D^{2}{\left(\mathrm{mm}\right)}^{2}}$$
where *D* is the average length of the diagonals, and *F* is the load applied by the indenter [11].

The use of machine learning techniques has increased in recent years, as they are employed to optimize, monitor, and control industrial processes. These algorithms possess the capability to learn and adapt to dynamic systems [4]. For instance, Martins et al. developed an algorithm to automatically inspect surface defects in rolled steel, classifying issues such as oxidation, exfoliation, and waveform defects using computer vision and neural networks, achieving an accuracy of 87%, ensuring high-quality steel production [6]. Furthermore, the study and determination of adhesion strength between the coating/substrate system are of great importance as they allow the estimation of the quality and the type of deformation exhibited by the coating. Bastian Lenz et al. utilized convolutional neural networks to classify the type of adhesion present in a set of Rockwell indentation images, showing promising results for automated industrial applications [5].

The comparison of machine learning techniques is of great importance to determine the approach that exhibits higher performance, and shorter execution time. Mohamad et al. predicted the hardness performance of TiAlN coatings using support vector machine (SVM), artificial neural network (ANN), and RSM-fuzzy. Parameters such as sputtering power, polarization voltage, and substrate temperature were used as input, where the SVM model showed better approximation to experimentally obtained hardness results compared to other study methods [8]. Wen et al. implemented an algorithm to search for high-entropy alloys with high hardness in an Al-Co-Cr-Cu-Fe-Ni system. They predicted material hardness based on composition and other descriptors (atomic radii, valence electrons, among others) using machine learning techniques such as linear regression, polynomial regression, support vector regression (SVR) with linear, polynomial, and radial kernels, regression tree, backpropagation artificial neural network (ANN), and k-nearest neighbors (KNN). SVR with a radial kernel showed lower prediction error compared to other techniques [9]. Waleed et al. implemented machine learning techniques such as multivariable linear regressor (LR), Gaussian process regressor (GPR), and SVR to predict properties of AA6061 compounds, including relative density, hardness, grain size, among others. The goal was to optimize input parameters such as the percentage of silicon carbide particles (SiCp), pressure for high torsion, and number of revolutions. The results indicated that SVR demonstrated better performance in predicting these properties [12]. Similarly, Keya Fu et al. utilized machine learning techniques such as random forest (RF), k-nearest neighbors (KNN), linear regression (LR), XGBoost, light gradient boosting machine (LightGBM), and an artificial neural network (ANN) to predict the tensile strength of aluminum alloys based on their elemental chemical composition and grain size. The XGBoost technique yielded better prediction results compared to other studied techniques [13].

On the other hand, researchers have implemented image processing methods and machine learning techniques for the determination of Vickers hardness. Polanco et al. estimated the Vickers hardness value using image processing methods to detect the corners of the indentation imprint, achieving a maximum error of 4.5% compared to manually obtained values [10]. Dovale et al. used a gradient boosting regressor (GBR) to predict hardness based on mechanical properties such as bulk modulus (B), shear modulus (G), Young’s modulus (Y), and Poisson’s ratio (ν). They also implemented the classification model (GBC) to predict the best relationship for calculating hardness with these input variables [14]. Jeon et al. employed support vector regression (SVR), k-nearest neighbors (kNN), random forest regression (RFR), and artificial neural networks (ANN) to predict the hardness of low-alloy steels under various tempering conditions such as temperature, holding time, and alloy composition. RFR showed the best $R^{2}$ value of 0.9966 compared to other machine learning models [15].

Swetlana et al. implemented image processing to extract microstructural descriptors such as phase distribution, volume fraction, number and size of particles, elemental chemical composition, temperature, and annealing time. These descriptors were used and combined in three different datasets to predict Vickers hardness using Gaussian process regression (GPR), achieving mean squared errors between 0.59 and 0.15 [16]. Privezentsev et al. developed software with indentation images that binarizes, filters, and calculates geometric characteristics of the image using artificial neural networks (ANN). They found that the differences in geometric parameters of the indentation imprint calculated manually compared to the software did not exceed 4% [17]. Tanaka et al. implemented two automatic methods based on convolutional neural networks (CNNs), using images with ideal surfaces, rough surfaces, distorted indentations, and cracks as inputs to measure indentation diagonals and determine Vickers hardness automatically and robustly. They reported errors between 0.1% and 6% in Vickers hardness value, indicating that their method allows for precision close to that of a human operator [18]. Buitrago et al. employed CNNs to determine the Vickers hardness of D2 steel and TiNbN coating by predicting the position of the corners of the indentation imprint, obtaining errors between 0.17% and 5.98% [19].

According to the studies presented previously, it can be observed that machine learning techniques are used to predict Vickers hardness based on material characteristics such as chemical composition, morphology, and material properties. Additionally, these techniques have been implemented to detect the indentation imprint or its corners, thereby determining the value of Vickers hardness.

In this work, a new method of predicting Vickers hardness is implemented without the need to measure the diagonals of the indentation imprint. This is achieved from image conditions such as scale, whether the analyzed image presents coating or not, applied force, and scanning of the indentation imprint image in grayscale. This study employs machine learning techniques such as Decision Tree (DT), AdaBoost, Extreme Gradient Boosting (XGB), and Random Forest (RF), with the aim of identifying the most suitable machine learning technique for predicting Vickers hardness.

## 2. Materials and Methods

### 2.1. Materials

Fifty-four Vickers indentation images of commercially available D2 steel, water-quenched from 1000 °C, tempered at 400 °C for 90 min, and TiNbN coatings deposited by Arc-PVD at varying substrate temperatures ($T_{s}=$ 200 °C, 400 °C and 600 °C) were analyzed. The indentation images of the coating, the Vickers hardness and the elemental chemical composition were obtained from previous studies [19,20]. This type of material is implemented as a cutting tool, which must have high hardness and wear resistance. Table 1 shows the chemical composition of the coating and the average Vickers hardness of the study materials using a load of 10 N.

Each image was obtained under a distinct applied load. These images were obtained using two optical microscopes: the OLYMPUS UPRIGHT BX51FM (OLYMPUS, Tokyo, Japan), equipped with a 10.2 Mpx SC100 camera utilizing CMOS technology, providing a magnification of 20 μm and dimensions of 3840 × 2748 px (see Figure 2a–c); and the OLYMPUS DSX500i, featuring an 18 Mpx CCD camera, offering a magnification of 10 μm and dimensions of 1194 × 1194 px (see Figure 2d–g). Rotations were performed every 5° from 0° to 360° and horizontal mirroring was applied to each rotation to increase the data diversity and reduce the effects of directional bias in the indentation images. This resulted in a total of 7776 images, which represents a 144-fold increase from the original database size of 54 images.

A database was constructed to predict Vickers hardness from indentation imprints. Descriptors include scale, coating presence (binary encoded), applied load (2, 3, 5, or 10 N), and grayscale indentation images. The output variable is Vickers hardness, calculated from the indentation image using established methods (see Table 2).

To determine the image size yielding the highest performance, and considering each pixel as a descriptor, the resizing process also aimed to expedite computation and ensure uniform image dimensions for optimal performance. Indentation images were therefore resized to relatively small dimensions of 10×10, 25×25, 50×50, 75×75, and 100×100 pixels. Decision trees were utilized, trained on 80% of the dataset and validated on the remaining 20%. To assess the performance of machine learning models, metrics such as RMSE, MAE and $R^{2}$ were used:Root Mean Square Error (RMSE): It is the square root of the Mean Square Error (MSE), measures the square of the differences between predicted and actual values (see Equation (Equation 2)).Mean Absolute Error (MAE): It is the average of the absolute differences between predicted and actual values (see Equation (Equation 3)).The coefficient of determination ($R^{2}$): Provides information on how well a model’s predictions fit the actual data (see Equation (Equation 4)). If $R^{2}≊$ 1, it indicates that the model explains or fits the data well; otherwise, if $R^{2}$ <= 0, it indicates that the model does not fit the data.
(2)$$RMSE=\sqrt{MSE}=\sqrt{\frac{1}{n}\sum_{i=1}^{n}{\left(y_{i}-{\hat{y}}_{i}\right)}^{2}}$$
(3)$$MAE=\frac{1}{n}\sum_{i=1}^{n}\left|y_{i}-{\hat{y}}_{i}\right|$$
(4)$$R^{2}=1-\frac{\sum_{i=1}^{n}{\left(y_{i}-{\hat{y}}_{i}\right)}^{2}}{\sum_{i=1}^{n}{\left(y_{i}-{\bar{y}}_{i}\right)}^{2}}$$
where *n* is the number of observations, $y_{i}$ is the actual value of the dependent variable, ${\hat{y}}_{i}$ is the predicted value by the model, and ${\bar{y}}_{i}$ is the mean value of the actual values.

The Table 3 summarizes the results, encompassing training time ($Time_{Train}$), validation time ($Time_{Test}$), root mean square error (RMSE), mean absolute error (MAE), and coefficient of determination ($R^{2}$). Notably, the highest performance was achieved when the image size was 50×50 pixels, indicating its suitability for this study (see Figure 3).

### 2.2. Methods

Developing performance and efficient methods for predicting Vickers hardness values is crucial for material science and engineering applications. Considering the small data size, we focused on algorithms known for their robustness and ability to handle smaller datasets. We specifically investigated four prominent techniques: Decision Tree (DT), Adaptive Boosting (ADABoost), Extreme Gradient Boosting (XGBoost), and Random Forest (RF).

**Decission Tree (DT):** DT is a supervised learning algorithm that builds classification and regression models using a hierarchical tree structure. It recursively partitions the dataset into smaller subsets based on specific features, eventually reaching leaf nodes with predicted target values. Key parameters for DT include maximum tree depth, which controls the complexity of the tree, and the splitting criterion, which determines the best feature and threshold for each split. Common criteria include Friedman Mean Squared Error (MSE), squared error, and absolute error [21,22].-Friedman MSE: This method utilizes the mean squared error with Friedman’s improvement score for potential splits.-Squared Error: The mean squared error serves as the feature selection criterion, aiming to minimize the L2 loss by assessing the reduction of variance at each terminal node.-Absolute Error: The mean absolute error minimizes the L1 loss by utilizing the median of each terminal node.**Adaptive Boosting or ADABoost:** It is a meta-estimator that incrementally grows in complexity with each boosting iteration. It employs small decision tree estimators as weak learners, which are added sequentially. Each subsequent model aims to correct the predictions of its predecessor, thereby enhancing overall predictive performance [23].**Extreme Gradient Boosting or XGBoost:** XGBoost is a method designed for improving Gradient Boosting. It utilizes a gradient descent algorithm to minimize the loss when adding new models. In regression tasks, XGBoost employs small decision trees, where each new tree predicts the residuals or errors of the previous trees. These predictions are then combined with the previous tree to make the final prediction [24].**Random Forest or RF:** Random Forest involves building prediction models in classification or regression from a set of decision trees without interaction between them. Key parameters include the criterion, which employs functions like Friedman MSE, squared error, and absolute error, the maximum depth of each tree, and the number of estimators, referring to the number of trees in the forest [25].

The dataset was split into training (60%), tuning (20%), and testing (20%) sets. Grid Search was employed to identify the optimal hyperparameter configuration for the chosen machine learning model. Cross-validation was performed using 5 folds, repeated 5 times. The average performance over the 5 folds was used to select the best hyperparameter configuration. The flowchart describing the aforementioned methodology is shown in Figure 4. Python 3.11.5 and scikit-learn were used for implementation on a computer with 12 GB RAM, an 11th Gen Intel^®^ Core™ i5-1155G7 CPU @ 2.50 GHz, and Windows 11.

## 3. Results and Discussion

### 3.1. Hyperparameters Tuning

For the hyperparameters fit of the different models studied, the criterion, maximum depth of the decision tree, and the number of estimators for random forests, ADABoost, and XGBoost were considered. Figure 5 depicts plots of the tuned hyperparameters for each machine learning technique with the best-selected values. Table 4 presents the obtained values.

### 3.2. Model Testing and Validation

Once the tuning hyperparameters were identified, the machine learning techniques were trained. The model with the best performance was then used for validation with the test set. Table 5 shows the results of training score (Train score), test time, Root Mean Square Error (RMSE), Mean Absolute Error (MAE), and $R^{2}$ for each model.

As shown in Table 5, the training score values for all models are close to 1, indicating successful learning. However, the validation results reveal that DT, XGBoost, and RF outperform ADABoost. This is because ADABoost uses decision trees with a maximum depth of 3. According to the DT tuning plots (see Figure 5), DT achieves better $R^{2}$ values for depths higher than 10. Consequently, ADABoost exhibits higher RMSE and MAE errors, suggesting that deeper parameter tuning is crucial for improved predictions.

While the $R^{2}$ values for DT, XGBoost, and RF are similar, the Random Forest (RF) model demonstrates the best fit in the test results, followed by XGBoost and DT. This is corroborated by its lower RMSE values, as RMSE is sensitive to outliers and penalizes significant discrepancies between predicted and actual values.

Figure 6 presents the prediction vs. true value plots for all machine learning techniques. It is evident that the predictions for DT, RF, and XGBoost align well with the true values compared to ADABoost.

The Random Forest machine learning technique demonstrated superior performance. This model was utilized to forecast the Vickers hardness value using 10 images not included in the database, with the objective of assessing the model’s efficacy in Vickers hardness value prediction. The error percentage for these images ranged from 0.43% to 6.88%, with an average execution time of 0.14 ± 0.10 s.

Previous studies employing image processing techniques, like those by Polanco et al. [10], reported hardness measurement errors ranging from 0.32% to 4.5% for both manual and their proposed methods, with an average processing time of 2.05 s. Similarly, Buitrago et al. [19] utilized convolutional neural networks (CNNs) and achieved manual hardness errors between 0.17% and 5.98%, with an average execution time of 6 s. Our proposed method demonstrates comparable error margins to these existing approaches, while significantly reducing processing time, thereby leading to lower computational costs.

As depicted in Table 6 and Table 7, as the indentation size increases significantly relative to the pore size and if background noise diminishes, the model’s performance improves, as evident in Table 6 and Table 7.

As observed, the results obtained with this novel technique, which circumvents the calculation of diagonal lengths to determine hardness value, are akin to those obtained with corner detection-based methods, thus opening new avenues for material characterization based on Vickers hardness.

Despite the limited number of images contained in the database and the low diversity of the study materials, results with an error rate of less than 6.88% were obtained. One way to improve these results in the future is to expand the database to enable better machine learning.

## 4. Conclusions

In this work, a new method based on machine learning techniques was developed to predict the Vickers hardness value from the indentation image, applied load, whether it has a coating or not, and the image scale. The method achieved low RMSE and MAE errors (095 and 0.12 respectively), with an $R^{2}$ close to 1 when employing the Random Forest technique.Evaluating the size of the image in the database is crucial since each pixel of the image acts as a descriptor. A larger image introduces redundant information or noise, leading to high RMSE and MAE errors and increasing computation time. Conversely, a very small image can result in a significant loss of information, making learning challenging. Therefore, an image size of 50 × 50 pixels reduces computation time and yields good results RMSE ≈0, MAE ≈0, and $R^{2}$≈1. This size minimizes information loss, facilitating effective data learning.In the metal-mechanic industry, material characterization is crucial. By determining the Vickers hardness value, it is possible to assess the quality of the material, whether it has undergone heat treatment, and if the coating is suitable for specific applications. This prevents the material from fracturing or deforming during short-term operation. Therefore, the proposed method charts a new course, diverging from traditional reliance on corner detection, for characterizing materials based on Vickers hardness, generating more efficient quality control and greater reliability of the final product.

## Figures and Tables

**Figure 1 materials-17-02235-f001:**
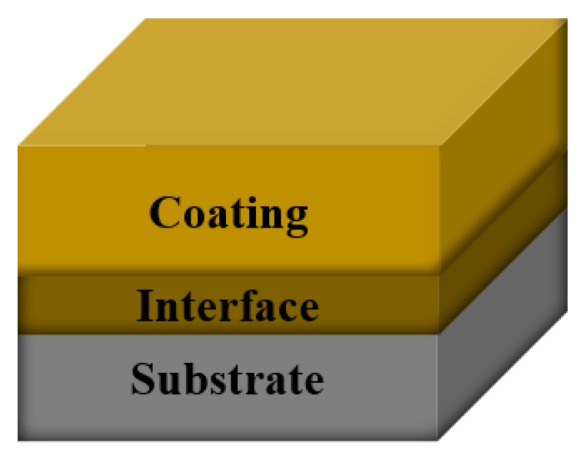
Thin film coating on substrate.

**Figure 2 materials-17-02235-f002:**
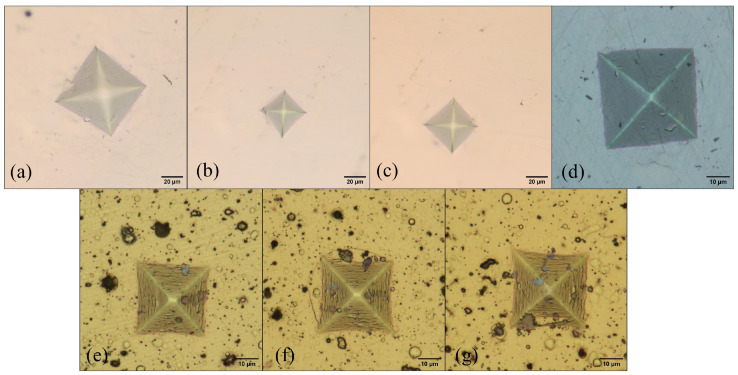
Indented images showcase various treatments of D2 Steel: (**a**) untreated, (**b**) quenched, (**c**,**d**) tempered, alongside TiNbN coating at different substrate temperatures (Ts): (**e**) Ts = 200 °C, (**f**) Ts = 400 °C, and (**g**) Ts = 600 °C.

**Figure 3 materials-17-02235-f003:**
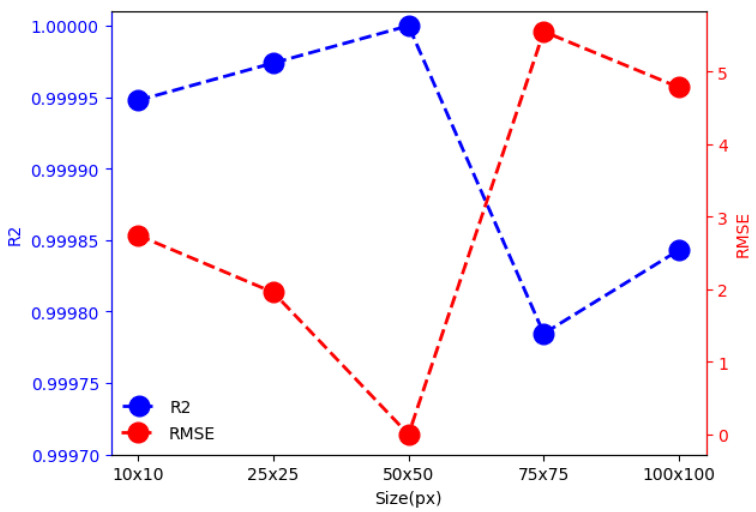
Effect of Image Size on Coefficient of Determination (R^2^) (Blue Line) and Root Mean Square Error (RMSE) (Red Line).

**Figure 4 materials-17-02235-f004:**
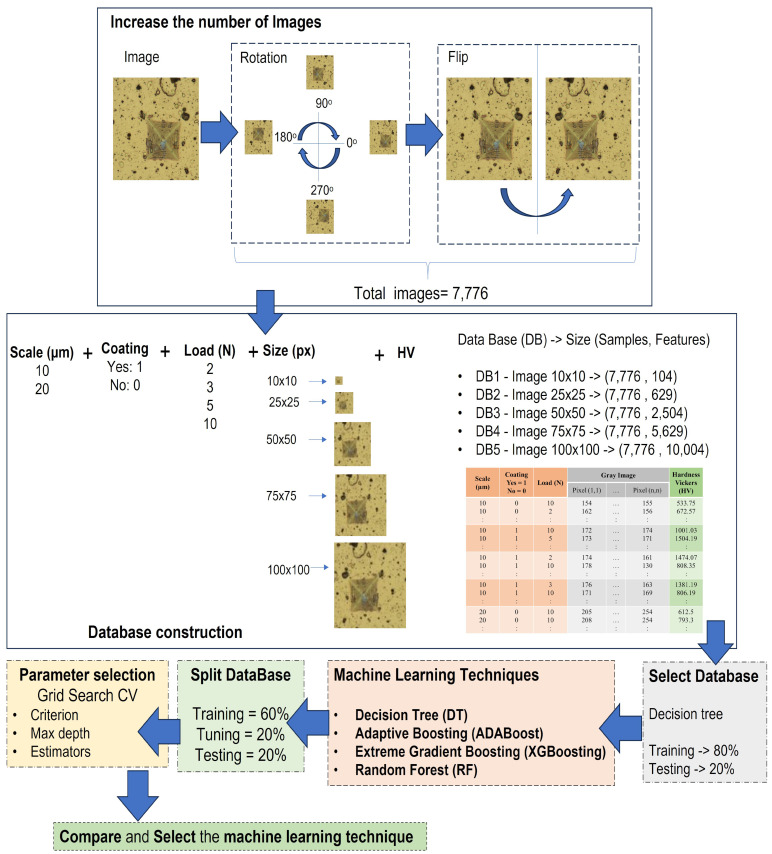
Flowchart of the proposed approach.

**Figure 5 materials-17-02235-f005:**
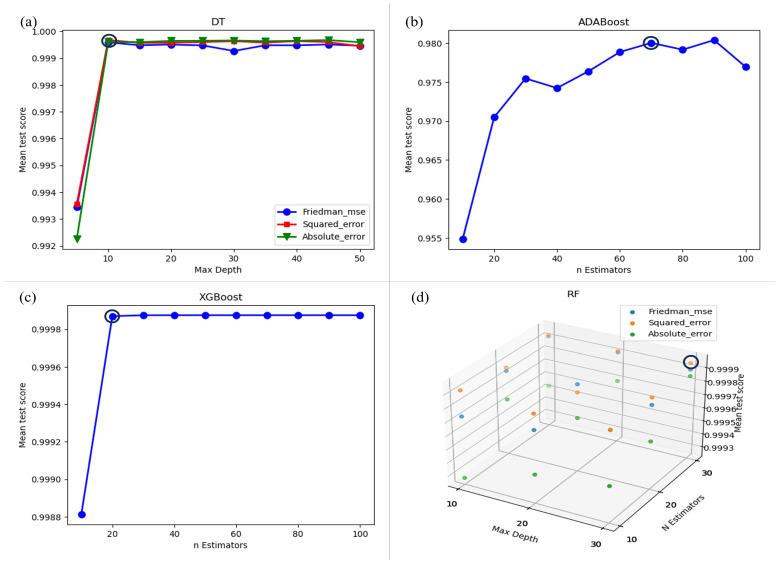
Tuned hyperparameters for each machine learning technique (**a**) DT, (**b**) ADABoost, (**c**) XGBoost and (**d**) RF.

**Figure 6 materials-17-02235-f006:**
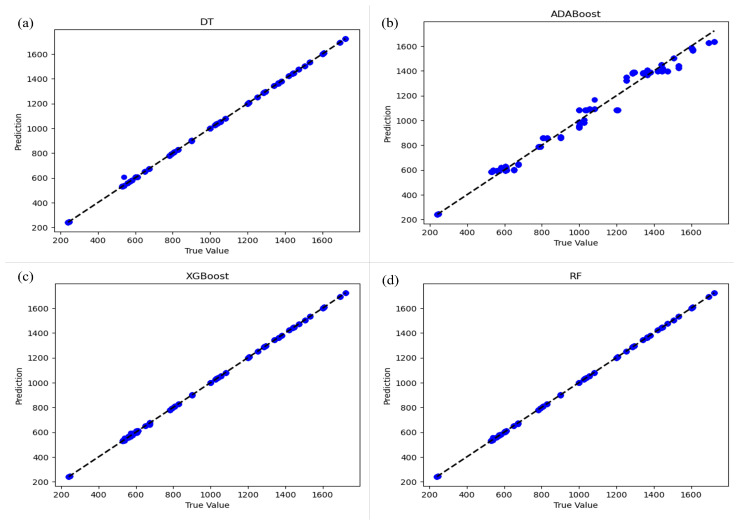
Prediction vs. true value plots for all machine learning techniques (**a**) DT, (**b**) ADABoost, (**c**) XGBoost and (**d**) RF.

**Table 1 materials-17-02235-t001:** Chemical composition of the TiNbN coating and average Vickers hardness of each material [19,20].

Material	Ti (at%)	Nb (at%)	N (at%)	${\mathrm{HV}}_{10N}$
D2 Steel				243.25 ± 2.45
D2 Quenched				787.05 ± 6.25
D2 Tempered				573.17 ± 55.72
TiNbN 200 °C	57.86 ± 7.28	0.21 ± 0.01	41.93 ± 7.27	1009.21 ± 14.53
TiNbN 400 °C	51.06 ± 0.57	0.18 ± 0.02	48.76 ± 0.58	846.24 ± 49.16
TiNbN 600 °C	44.72 ± 3.09	0.15 ± 0.02	55.12 ± 3.08	844.35 ± 49.02

**Table 2 materials-17-02235-t002:** Descriptors and Hardness measures included in the database.

Scale (μm)	Coating Yes = 1 No = 0	Load (N)	Gray Image Pixel (1,1) … Pixel (n,n)		Hardness Vickers (HV ±0.05)
10 10	0 0	10 2	154 162	155 156	533.75 672.57
10 10	1 1	10 5	172 173	174 171	1001.03 1504.19
10 10	1 1	2 10	174 178	161 130	1474.07 808.35
10 10	1 1	3 10	176 171	163 169	1381.19 806.19
20 20	0 0	10 10	205 208	254 254	612.50 793.30

**Table 3 materials-17-02235-t003:** Time, root mean squared error (RMSE), mean absolute error (MAE) and coefficient of determination results obtained with the five databases constructed by changing the input image size.

Image Size (px)	${\mathrm{Time}}_{\mathrm{Train}}$ (s)	${\mathrm{Time}}_{\mathrm{Test}}$ (s)	RMSE	MAE	$R^{2}$
10 × 10	0.19	0.00	2.74	0.10	0.99995
25 × 25	1.13	0.00	1.96	0.09	0.99997
50 × 50	4.70	0.02	1.54$\times 10^{-12}$	1.15 $\times \;10^{-12}$	1.00
75 × 75	11.17	0.02	5.55	0.27	0.99978
100 × 100	20.68	0.03	4.78	0.29	0.99984

**Table 4 materials-17-02235-t004:** Average score for selected tuning hyperparameters.

ML ^1^	Criterion	Max Depth	Estimators	Mean Score
DT	Squared Error	10		0.9997
ADABoost			70	0.9801
XGBoost			20	0.9999
RF	Squared Error	30	30	0.9999

^1^ ML: Machine Learning.

**Table 5 materials-17-02235-t005:** Results of train score, test time, $R^{2}$, MAE, and RMSE for each model.

ML	Train Score	Times Test (s)	RMSE	MAE	$R^{2}$
DT	1.00	0.02	1.94	0.21	0.99997
ADABoost	0.98	1.17	53.10	43.19	0.98085
XGBoost	1.00	0.02	1.71	0.71	0.99998
RF	1.00	0.06	0.95	0.12	0.99999

**Table 6 materials-17-02235-t006:** Vickers hardness prediction of coating using random forest.

Image	Scale (μm)	Coating	Load (N)	HV True	HV Predict	%Error
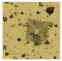	10	1	10	931.31	992.93	6.62
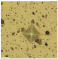	10	1	5	1108.42	1100.93	0.68
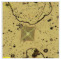	10	1	5	1104.34	1099.42	0.45
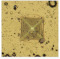	10	1	10	801.35	838.91	4.69
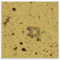	10	1	3	1269.00	1356.37	6.88

**Table 7 materials-17-02235-t007:** Vickers hardness prediction of steel using random forest.

Image	Scale (μm)	Coating	Load (N)	HV True	HV Predict	%Error
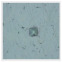	10	1	10	931.31	992.93	6.62
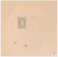	10	1	5	1108.42	1100.93	0.68
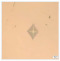	10	1	5	1104.34	1099.42	0.45
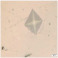	10	1	10	801.35	838.91	4.69
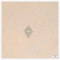	10	1	3	1269.00	1356.37	6.88

## Data Availability

The data presented in this study are available upon request to the corresponding author. The data are not available to the public because they are preliminary results of an ongoing research project carried out in collaboration between CONAHCYT and the Universidad de Ibagué. Furthermore, this information will be used in future technological developments and will be subject to intellectual property protection.

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
