# Peer review of "Predictive Modeling of Vickers Hardness Using Machine Learning Techniques on D2 Steel with Various Treatments"

_materials, 2024, doi:10.3390/ma17102235_

Round 1

Reviewer 1 Report

Comments and Suggestions for Authors

In this paper, regression techniques such as decision tree, adaptive enhancement, extreme gradient enhancement and random forest are used to predict Vickers hardness by scanning indentation monochromatic images without measuring their diagonal lines. Compared with the techniques studied, the random forest technique has obtained good results in terms of indicators, the root mean square error (RMSE) is 0.95, the absolute mean error (MAE) is 0.12, and the measurement coefficient (R2) is about 1. This study has certain research value and is worth publishing, but there are some deficiencies that need to be revised.

1. The capital letter abbreviation that appears for the first time in the abstract needs to be written in its full name, otherwise the specific meaning is difficult for the reader, and it is recommended to modify;

2. Please add the last paragraph in the Introduction to summarize the differences between this paper and previous literature studies, so as to reflect the value of this study. Please revise it.

3. In the Materials section, relevant parameters such as the composition and performance of TiNbN materials should be provided, which is the basis of machine learning technology, please add;

4. The title of Figure 3 is too simple, please be more specific;

5. In the conclusion part, all are qualitative conclusions, there is no specific conclusion data, please modify; In addition, please follow (1), (2), (3)...... To summarize the conclusions of the full text, to highlight the key points, please revise;

6. The table samples in the paper are inconsistent, such as Table 1 and Table 2, please modify them. Figure 6 and Figure 7 are actually tables, but the author thinks they are graphs, please check and confirm to make them consistent.

Author Response

Dear Reviewer,

We express our gratitude for the feedback from the editor and reviewers. We have meticulously revised the original manuscript, considering all the suggestions and comments received, which we have incorporated into the updated version of the document. We believe that the contributions from the editors and reviewers have been crucial in significantly improving the manuscript. We trust that the adjustments made and the detailed explanations provided in this document adequately address your concerns. We hope that the modified manuscript will represent a valuable contribution for interested readers. Below, we present our detailed responses.

  1. Reviewer’s comment: The capital letter abbreviation that appears for the first time in the abstract needs to be written in its full name, otherwise the specific meaning is difficult for the reader, and it is recommended to modify.

Author’s response: The respective modification of the uppercase abbreviation was made on lines 13 and 14 of the abstract: “Root Mean Square Error (RMSE) of 0.95, Mean Absolute Error (MAE) of 0.12, and Coefficient of Determination (R2) ≈ 1”

  1. Reviewer’s comment: Please add the last paragraph in the Introduction to summarize the differences between this paper and previous literature studies, so as to reflect the value of this study. Please revise it.

Author’s response: Two paragraphs were added at the end of the introduction (lines 110 – 122).

“According to the studies presented previously, it can be observed that machine learning techniques are used to predict Vickers hardness based on material characteristics such as chemical composition, morphology, and material properties. Additionally, these techniques have been implemented to detect the indentation imprint or its corners, thereby determining the value of Vickers hardness.

In this work, a new method of predicting Vickers hardness is implemented without the need to measure the diagonals of the indentation imprint. This is achieved from image conditions such as scale, whether the analyzed image presents coating or not, applied force, and scanning of the indentation imprint image in grayscale. This study employs machine learning techniques such as Decision Tree (DT), AdaBoost, Extreme Gradient Boosting (XGB), and Random Forest (RF), with the aim of identifying the most suitable machine learning technique for predicting Vickers hardness.”

  1. Reviewer’s comment: In the Materials section, relevant parameters such as the composition and performance of TiNbN materials should be provided, which is the basis of machine learning technology, please add.

Author’s response: Information was provided on the elemental chemical composition of the coatings and the Vickers hardness with a load of 10 N for all study materials (lines 127 – 132).

“TiNbN coatings deposited by Arc-PVD at varying substrate temperatures (Ts = 200°C, 400°C and 600°C) were analyzed. The indentation images of the coating, the Vickers hardness and the elemental chemical composition were obtained from previous studies [19, 20]. This type of material is implemented as a cutting tool, which must have high hardness and wear resistance. Table 1 shows the chemical composition of the coating and the average Vickers hardness of the study materials using a load of 10N.”

  1. Reviewer’s comment: The title of Figure 3 is too simple, please be more specific.

Author’s response: The following modification was made to the title of the image, describing what it represents:” Flowchart of the proposed approach.” (page 8).

  1. Reviewer’s comment: In the conclusion part, all are qualitative conclusions, there is no specific conclusion data, please modify; In addition, please follow (1), (2), (3)...... To summarize the conclusions of the full text, to highlight the key points, please revise.

Author’s response: Numerical enumerations were added to the conclusions, as well as the numerical values of the RMSE and MAE results. One conclusion was modified, and finally, an additional conclusion was added:

  1. In this work, a new method based on machine learning techniques was developed to predict the Vickers hardness value from the indentation image, applied load, whether it has a coating or not, and the image scale. The method achieved low RMSE and MAE errors (095 and 0.12 respectively), with an R2 close to 1 when employing the Random Forest technique.

  1. Evaluating the size of the image in the database is crucial since each pixel of the image acts as a descriptor. A larger image introduces redundant information or noise, leading to high RMSE and MAE errors and increasing computation time. Conversely, a very small image can result in a significant loss of information, making learning challenging. Therefore, an image size of 50x50 pixels reduces computation time and yields good results RMSE ≈ 0, MAE ≈ 0, and R2 ≈ 1. This size minimizes information loss, facilitating effective data learning.

  1. In the metal-mechanic industry, material characterization is crucial. By determining the Vickers hardness value, it is possible to assess the quality of the material, whether it has undergone heat treatment, and if the coating is suitable for specific applications. This prevents the material from fracturing or deforming during short-term operation. Therefore, the proposed method charts a new course, diverging from traditional reliance on corner detection, for characterizing materials based on Vickers hardness, generating more efficient quality control and greater reliability of the final product.

  1. Reviewer’s comment: The table samples in the paper are inconsistent, such as Table 1 and Table 2, please modify them. Figure 6 and Figure 7 are actually tables, but the author thinks they are graphs, please check and confirm to make them consistent.

Author’s response: In accordance with your suggestions, the captions were modified to enhance the clarity of the content in Tables 1 and 2. These tables were subsequently renumbered as Tables 2 and 3.

Table 2. Descriptors and Hardness measures included in the database.

Table 3. Time, root mean squared error (RMSE), mean absolute error (MAE) and coefficient of determination results obtained with the five databases constructed by changing the input image size.

The labeling of Figures 6 and 7 was modified to Tables 5 and 6.

Reviewer 2 Report

Comments and Suggestions for Authors

comment in the attachment

Author Response

Dear Reviewer,

We express our gratitude for the feedback from the editor and reviewers. We have meticulously revised the original manuscript, considering all the suggestions and comments received, which we have incorporated into the updated version of the document. We believe that the contributions from the editors and reviewers have been crucial in significantly improving the manuscript. We trust that the adjustments made and the detailed explanations provided in this document adequately address your concerns. We hope that the modified manuscript will represent a valuable contribution for interested readers. Below, we present our detailed responses.

  1. Reviewer’s comment: The abstract should also provide some practical conclusions to increase reader interest.

Author’s response: This text was added to the abstract: ‘These results suggest that employing machine learning algorithms for predicting Vickers hardness from scanned images offers a promising avenue for rapid and accurate material assessment, potentially streamlining quality control processes in industrial settings.’

  1. Reviewer’s comment: Considering the fact that any research must have an end in industry, please propose a practical conclusion by which specialists in the field can improve their productivity using your research.

Author’s response: In the metal-mechanic industry, material characterization is crucial. By determining the Vickers hardness value, it is possible to assess the quality of the material, whether it has undergone heat treatment, and if the coating is suitable for specific applications. This prevents the material from fracturing or deforming during short-term operation. Therefore, the proposed method charts a new course, diverging from traditional reliance on corner detection, for characterizing materials based on Vickers hardness, generating more efficient quality control and greater reliability of the final product.

  1. Reviewer’s comment: How did the author ensure the accuracy of the proposed method?

Author’s response: We appreciate your feedback and concur with your observation. Indeed, in the context of regression methods, the concepts of accuracy and precision are not applicable. Instead, appropriate error metrics such as RMSE, MAE, and R2 are employed.

  1. Reviewer’s comment: Image preprocessing generally comprises spatial domain methods and frequency domain methods. The main preprocessing algorithms include grayscale transformation, histogram equalization, various filtering algorithms based on spatial and frequency domains, etc. In addition, mathematical morphology can also be used for image denoising. Why didn't the authors use image analysis (e.g. filtering, edge detection...)?

Author’s response: We did not use image analysis techniques such as filtering or edge detection because our focus was specifically on evaluating the effectiveness of machine learning algorithms in predicting Vickers hardness directly from grayscale images of indentation prints. In previous works, we have employed image processing techniques including filters, mathematical morphology, etc., as well as convolutional neural networks to detect indentation corners.

  1. Reviewer’s comment: What is the influence of the type and mode of lighting on the obtained research results?

Author’s response: Considering that the intensity of illumination varies according to the image's tonality, lighting does indeed have an effect on hardness prediction. Therefore, analyzing images by varying the type and mode of illumination could be considered as a new perspective, as it can expand the database by implementing images with varying brightness.

  1. Reviewer’s comment: CCD or CMOS image sensor technology is essential for image capturing. They convert optical signals into electrical signals. However, these two types of chips adopt different methods and means in the transmission of this information and their respective designs are totally different. Which technology did the authors use in their research? Please justify;

Author’s response: The images were acquired using two types of microscopes:

  • OLYMPUS UPRIGHT BX51FM, which uses a 10.2 Mpx SC100 camera with CMOS technology.
  • OLYMPUS DSX500i, which uses an 18Mpx camera with CCD technology.

This information has been added to the article in the materials section.

These images were obtained using two optical microscopes: the OLYMPUS UPRIGHT BX51FM, equipped with a 10.2 Mpx SC100 camera utilizing CMOS technology, providing a magnification of 20 μm and dimensions of 3840 x 2748 px (see Figure 2 a, b, and c); and the OLYMPUS DSX500i, featuring an 18Mpx CCD camera, offering a magnification of 10 μm and dimensions of 1194 x 1194 px (see Figure 2 d, e, f, and g)

  1. Reviewer’s comment: I propose to compare some traditional methods for feature extraction and hardness classification;

Author’s response: In order to evaluate the proposed technique against traditional methods, we have compared the error percentage of the results obtained by other authors implementing image processing techniques and convolutional neural networks. “Previous studies employing image processing techniques, like those by Polanco et al. reported hardness measurement errors ranging from 0.32\% to 4.5\% for both manual and their proposed methods, with an average processing time of 2.05 seconds. Similarly, Buitrago et al. utilized convolutional neural networks (CNNs) and achieved manual hardness errors between 0.17\% and 5.98\%, with an average execution time of 6 seconds. Our proposed method demonstrates comparable error margins to these existing approaches, while significantly reducing processing time, thereby leading to lower computational costs.”

  1. Reviewer’s comment: What are the limitations of the proposed method?

Author’s response: The limitation of the proposed method was added in the conclusions.

Despite the limited number of images contained in the database and the low diversity of the study materials, results with an error rate of less than 6.88% were obtained. One way to improve these results in the future is to expand the database to enable better machine learning.

Reviewer 3 Report

Comments and Suggestions for Authors

This paper describes the machine learning attempt to evaluate Vickers hardness measurement images.  They report the algorithm used and conclude that the pixels size of 50x50 is optimal in view of the balance of accuracy and the computational resources.  The random forest was found to give the best performance.  This is a simple work but useful for the readers.  It is desirable to add some reference about the optimal image sizes -- quantitative discussion on why 50 x 50 is optimal (Some trade-off curve, x-axis shows the image size, y1-axis shows computation time/power, and y2 axis shows accuracy ,  if possible ).

Author Response

Dear Reviewer,

We express our gratitude for the feedback from the editor and reviewers. We have meticulously revised the original manuscript, considering all the suggestions and comments received, which we have incorporated into the updated version of the document. We believe that the contributions from the editors and reviewers have been crucial in significantly improving the manuscript. We trust that the adjustments made and the detailed explanations provided in this document adequately address your concerns. We hope that the modified manuscript will represent a valuable contribution for interested readers. Below, we present our detailed responses.

Reviewer’s comment: This is a simple work but useful for the readers.  It is desirable to add some reference about the optimal image sizes -- quantitative discussion on why 50 x 50 is optimal (Some trade-off curve, x-axis shows the image size, y1-axis shows computation time/power, and y2 axis shows accuracy ,  if possible ).

Author’s response: Taking your comments into consideration, in order for the reader to visualize the results related to the image size, the following graph has been added.
